# O-MMGP: Optimal Mesh Morphing Gaussian Process Regression for Solving PDEs with non-Parametric Geometric Variations

## Abstract

We address the computational challenges of solving parametric PDEs with non parametrized geometric variations and non-reducible problems, such as those involving shocks and discontinuities of variable positions. Traditional dimensionality reduction methods like POD struggle with these scenarios due to slowly decaying Kolmogorov widths. To overcome this, we propose a novel non-linear dimensionality reduction technique to reduce the required modes for representation. The non-linear reduction is obtained through a POD after applying a transformation on the fields, which we call *optimal mappings*, and is a solution to an optimization problem in infinite dimension. The proposed learning framework combines morphing techniques, non-linear dimensionality reduction, and Gaussian Process Regression (GPR). The problem is reformulated on a reference geometry before applying the dimensionality reduction. Our method learns both the optimal mapping, and the solution fields, using a series of GPR models, enabling efficient and accurate modeling of complex parametric PDEs posed on varibale geometries that share a common topology. The results obtained concur with current state-of-the-art models. We mainly compare our method with the winning solution of the ML4CFD NeurIPS 2024 competition.

## 1 Introduction

### 1.1 Background

Many scientific and engineering challenges involve solving complex boundary value problems, often formulated as parametric partial differential equations (PDEs). These problems require exploring the influence of varying parameters such as material properties, boundary and initial conditions, or geometric configurations. Traditional numerical methods like the finite element method and finite difference methods, while accurate, are computationally expensive, particularly when repeated evaluations are necessary across a large and high-dimensional parameter set. To address this computational burden, techniques in model-order reduction and machine learning have been developed, offering efficient approximations without compromising accuracy.

Model-order reduction techniques, such as the reduced-basis method [25, 14], construct low-dimensional approximation vector spaces to represent the set of solutions of the parametric PDE, enabling fast computation for new parameter values. These approaches typically involve an offline phase, where high-fidelity models are used to generate a reduced basis, and an online phase, where this basis is employed for efficient computations. However, when the physical domain varies with the parameters, standard methods like Proper Orthogonal Decomposition (POD) face challenges due to the need to reconcile solutions defined on different domains. This often requires morphing

techniques to map variable domains to a common one, transforming the problem into a form suitable for reduced-order modeling [21, 28]. Linear reduced-order modeling techniques also face huge difficulties when the solution set of interest has a slowly decaying Kolmogorov width. Such problems, which we call hereafter non-reducible problems, are common for PDEs that involve shocks of variable position, discontinuities, boundary layers an so on.

On the other hand, machine learning methods, particularly those relying on deep learning, have shown promise in solving PDEs, learning solutions, and accelerating computations [13, 19]. Approaches relying on graph neural networks (GNNs) [29] have been particularly effective in handling unstructured meshes and varying geometric configurations. Despite their flexibility, these methods require substantial computational resources and large training datasets, and they often lack robust predictive uncertainty estimates.

## 1.2 Contribution

In this work, we propose a novel approach that integrates principles of morphing, non-linear dimensionality reduction, and classical machine learning to address the challenges posed by parametric PDEs with geometric variations for non-reducible problems. The main contribution is a novel algorithm used to maximizes the energy in the first principal modes, which directly addresses the core limitation of traditional dimensionality reduction techniques, and limits the number of modes needed to approximate non-reducible problems in moderate dimension. Such approaches are commonly called registration in the literature. Unlike most existing algorithms that aim at projecting the samples on one single mode, our approach may involve an arbitrary number of modes $r \in \mathbb{N}^*$. In addition, another major advantage with respect to similar methods is that the procedure is done automatically as we do use any feature tracking and detection method.

## 1.3 Related works

The challenges mentioned in this work have been extensively studied in the literature. Morphing techniques have been used for various applications in reduced order modeling to recast the problem on a reference domain [24, 1, 35]. Numerous approaches were also proposed to deal with non-reducible problems by using non-linear dimensionality reduction such as registration [30, 32], feature tracking [22], optimal transport [9], neural networks [16, 12, 3, 5], and so on. Machine learning approaches leveraging neural networks have also demonstrated remarkable success in solving numerical simulations, particularly in capturing complex patterns and dynamics. Methods such as the Fourier Neural Operator (FNO) [18] and its extension, Geo-FNO [17], efficiently learn mappings between function spaces by leveraging Fourier transforms for high-dimensional problems. Physics-Informed Neural Networks (PINNs) [26] embed physical laws directly into the loss function, enabling solutions that adhere to governing equations. Deep Operator Networks (DeepONets) [20] excel in learning operators with small data requirements, offering flexibility in various applications. Mesh Graph Networks (MGNs) [23] use graph-based representations to model simulations on irregular domains, preserving geometric and topological properties.

## 2 Preliminaries and notations

Let $n \in \mathbb{N}^*$ and $\Omega_1, \ldots, \Omega_n \subset \mathbb{R}^d$ to be a family of $n$ distinct domains that share a common topology, with $d = 2, 3$. We suppose the parametrization of the domains is unknown and that for all $1 \leq i \leq n$, each domain $\Omega_i$ is equipped with a (non-geometrical) parameter $\mu_i \in \mathcal{P}$ where $\mathcal{P} \subset \mathbb{R}^p$ is a set of parameter values (think of $\mu_i$ as being some material parameter value for instance). In addition, let $u_i : \Omega_i \to \mathbb{R}$ be the solution of a parametrized partial differential equation for the parameter value $\mu_i$ defined on $\Omega_i$ (think about a temperature field for instance). We assume in the following that for all $1 \leq i \leq n$, $u_i \in L^2(\Omega_i)$. We assume that for all $1 \leq i \leq n$ a finite element mesh $\mathcal{M}_i$ is chosen so that $\partial \mathcal{M}_i$ can be considered as an accurate enough approximation of the boundary of the domain $\partial \Omega_i$. We also assume the field $u_i$ can be accurately approximated by its finite element interpolation associated to the mesh $\mathcal{M}_i$. We fix a reference domain, denoted by $\Omega_0$, equipped with a mesh denoted by $\mathcal{M}_0$, that shares the same topology as the other domains. For all $0 \leq i \leq n$, $\mathbf{M}_i$ be the space of bijective $W^{1,\infty}$ mappings from $\Omega_0$ onto $\Omega_i$, so that for all $\boldsymbol{\psi} \in \mathbf{M}_i$, $\boldsymbol{\psi}(\Omega_0) = \Omega_i$. We also introduce $\mathbf{M} := \mathbf{M}_1 \times \mathbf{M}_2 \times \cdots \times \mathbf{M}_n$.

For any family of functions $\mathcal{H} = (h_i)_{1 \leq i \leq n} \in L^2(\Omega_0)^n$, we define the correlation matrix application $C_{\mathcal{H}}$ of $\mathcal{H}$ as the map

$$C_{\mathcal{H}} : \quad \mathbf{M}_0^n \to \mathcal{S}_n^+$$
$$\Phi \to C_{\mathcal{H}}[\Phi] := (C_{\mathcal{H},ij}[\Phi])_{1 \leq i,j \leq n}, \tag{1}$$

where $\mathcal{S}_n^+$ is the set of symmetric non-negative semi-definite matrices of dimension $n \times n$, and for all $\Phi := (\phi_i)_{1 \leq i \leq n} \in \mathbf{M}_0^n$ and for all $1 \leq i, j \leq n$,

$$C_{\mathcal{H},ij}[\Phi] := \langle h_i \circ \phi_i, h_j \circ \phi_j \rangle_{L^2(\Omega_0)} = \int_{\Omega_0} h_i \circ \phi_i(x) h_j \circ \phi_j(x) dx.$$

We denote by $\lambda_1^{\mathcal{H}}[\Phi] \geq \lambda_2^{\mathcal{H}}[\Phi] \geq \cdots \geq \lambda_n^{\mathcal{H}}[\Phi] \geq 0$ the eigenvalues of $C_{\mathcal{H}}[\Phi]$. We also denote by $(\zeta_1^{\mathcal{H}}[\Phi], \zeta_2^{\mathcal{H}}[\Phi], \cdots, \zeta_n^{\mathcal{H}}[\Phi]) \subset \mathbb{R}^n$ an orthonormal family of corresponding eigenvectors. Finally, given a positive integer $r \in \mathbb{N} \setminus \{0\}$, we define the functional

$$J_{\mathcal{H},r} : \quad \mathbf{M}_0^n \to \mathbb{R}$$

$$\Phi \mapsto J_{\mathcal{H},r}[\Phi] := \frac{\sum_{j=1}^{r} \lambda_j^{\mathcal{H}}[\Phi]}{Tr(C_{\mathcal{H}}[\Phi])}. \tag{2}$$

We now introduce the terminology of three different configurations that will be used throughout the paper.

1. First, we refer to the **physical configuration** as the collection of pairs $\{(\Omega_i, u_i)\}_{1 \leq i \leq n}$.

2. Given a reference domain $\Omega_0$ and $\Phi^{\text{geo}} := (\phi_i^{\text{geo}})_{1 \leq i \leq n} \in \mathbf{M}$, we refer to the **reference configuration** as the collection of pairs $\{(\Omega_0, u_i \circ \phi_i^{\text{geo}})\}_{1 \leq i \leq n}$. In this configuration, all the fields $u_i^{\text{ref}} := u_i \circ \phi_i^{\text{geo}}$ belong to $L^2(\Omega_0)$, and classical dimensionality reduction techniques such as PCA can be applied on the family $\mathcal{U} := (u_i^{\text{ref}})_{1 \leq i \leq n} \in L^2(\Omega_0)^n$.

3. Given the reference configuration and some $r \in \mathbb{N} \setminus \{0\}$, we refer to the **r-optimal configuration** as the collection of pairs $\{(\Omega_0, u_i^{\text{ref}} \circ \phi_{i,r}^{\text{opt}})\}_{1 \leq i \leq n}$, where $\Phi_r^{\text{opt}} := (\phi_{i,r}^{\text{opt}})_{1 \leq i \leq n} \in \mathbf{M}_0^n$ is a solution to the following maximization problem:

$$\text{find } \Phi_r^{\text{opt}} \in \arg \max_{\Phi \in \mathbf{M}_0^n} J_{\mathcal{U},r}[\Phi]. \tag{3}$$

The maximization problem (3) is considered so that the family $(u_i^{\text{opt}})_{1 \leq i \leq n} := (u_i^{\text{ref}} \circ \phi_{r,i}^{\text{opt}})$ can be accurately approximated by elements of a $r$-dimensional vector space.

# 3 Methodology

## 3.1 Training phase

In the training phase, we suppose that we have access to the dataset of triplets $\{(\Omega_i, \mu_i, u_i)\}_{1 \leq i \leq n}$. The domain $\Omega_i$ (or its mesh) and the parameter $\mu_i$ are the inputs to the physical solver, and the field $u_i$ is its output. We chose a reference domain (which can be one from the dataset) that shares the same topology.

### 3.1.1 Pretreatment

We perform these pretreatment steps in the training phase.

1. We pass from the physical configuration to the reference configuration (4) by computing for all $1 \leq i \leq n$, a mapping $\phi_i^{\text{geo}} \in \mathbf{M}_i$. We apply POD on the family $(\phi_i^{\text{geo}} - \boldsymbol{Id})_{1 \leq i \leq n}$ to obtain the POD modes $\{\zeta_i^{\text{geo}}\}_{1 \leq i \leq n}$ and the generalized coordinates $\{\alpha^i\}_{1 \leq i \leq n}$ where $\alpha^i = (\alpha_j^i)_{1 \leq j \leq s} \in \mathbb{R}^s$, such that

$$\forall 1 \leq j \leq s, \ \alpha_j^i = \langle \phi_i^{\text{geo}} - \boldsymbol{Id}, \zeta_j^{\text{geo}} \rangle_{\boldsymbol{L}^2(\Omega_0)},$$

and $s$ is the number of retained modes for the geometrical mappings. Each domain $\Omega_i$ is defined now by the vector $\alpha^i$.

2. We pass from the reference configuration to the optimal configuration (5) by solving problem 3. Once we obtain the functions $(\boldsymbol{\phi}_i^{\mathrm{opt}} - \boldsymbol{Id})_{1 \leq i \leq n}$, we apply POD to obtain the POD modes $\{\boldsymbol{\zeta}_i^{\mathrm{opt}}\}_{1 \leq i \leq n}$ and the generalized coordinates $\{\beta^i\}_{1 \leq i \leq n}$ where $\beta^i = \left(\beta_j^i\right)_{1 \leq j \leq t} \in \mathbb{R}^t$, such that

$$\forall 1 \leq j \leq t, \ \beta_j^i = \langle \boldsymbol{\phi}_i^{\mathrm{opt}} - \boldsymbol{Id}, \boldsymbol{\zeta}_j^{\mathrm{opt}} \rangle_{\boldsymbol{L}^2(\Omega_0)},$$

113 and $t$ is the number of retained modes for the optimal mappings.

3. Finally, after evaluating $\{u_i^{\mathrm{opt}}\}_{1 \leq i \leq n}$, we apply POD to obtain the POD modes $\{\psi_i\}_{1 \leq i \leq n}$ and the generalized coordinates $\{\gamma^i\}_{1 \leq i \leq n}$ where $\gamma^i = \left(\gamma_j^i\right)_{1 \leq j \leq r} \in \mathbb{R}^r$, such that

$$\forall 1 \leq j \leq r, \ \gamma_j^i = \langle \psi_i, u_j^{\mathrm{opt}} \rangle_{\boldsymbol{L}^2(\Omega_0)},$$

114 After applying these dimensionality reductions, we obtain the following three approximations:

115 $$\boldsymbol{\phi}_i^{\mathrm{ref}} \approx \boldsymbol{Id} + \sum_{j=1}^{s} \alpha_j^i \boldsymbol{\zeta}_j^{\mathrm{ref}}, \boldsymbol{\phi}_i^{\mathrm{opt}} \approx \boldsymbol{Id} + \sum_{j=1}^{t} \beta_j^i \boldsymbol{\zeta}_j^{\mathrm{opt}}, \text{ and } u_i^{\mathrm{opt}} \approx \sum_{j=1}^{r} \gamma_j^i \psi_j.$$

116 Notice that the passing from the physical configuration to the reference configuration is decoupled
117 and purely geometrical, that is it does not depends on the fields $\{u_i\}_{1 \leq i \leq n}$. For example, we show in
118 Figure 1 two forms of airfoils. $\boldsymbol{\phi}^{\mathrm{geo}}$ will transform one airfoil onto the other. On the other hand, the
119 transition to the optimal configuration is coupled between all the samples in the training dataset.

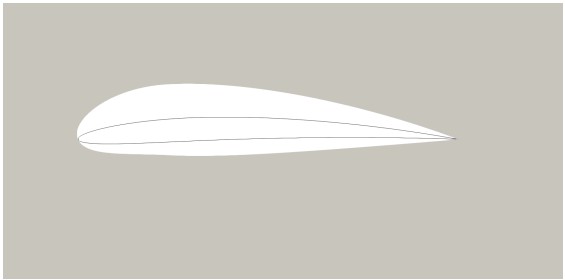

Figure 1: Two airfoils superimposed.

### 3.1.2 Training

121 After performing the dimensionality reduction step, we train two Gaussian processes regression
122 models [33] as follows.

1. The first model is to learn the optimal mapping that transforms the reference configuration to the optimal configuration. This model takes as input the physical parameter $\mu_i$ and the geometrical mapping POD coefficient $\alpha_i$, and as output the optimal mapping POD coefficient $\beta_i$. We denote this model by $\mathcal{R} : \mathbb{R}^p \times \mathbb{R}^q \to \mathbb{R}^t$.

2. The second model is to learn the field in the optimal configuration. This model takes as input the physical parameter $\mu_i$ and the geometrical mapping POD coefficient $\alpha_i$, and as output the field in the optimal configuration POD coefficient $\gamma_i$. We denote this model by $\mathcal{O} : \mathbb{R}^p \times \mathbb{R}^q \to \mathbb{R}^r$.

### 3.2 Inference phase

132 In the inference phase, we are given a new unseen geometry $\widetilde{\Omega}$ in the inference phase, with a physical
133 parameter $\widetilde{\mu}$. The goal is to predict the field of interest $\widetilde{u}$, solution to the physical simulation. We
134 proceed in the following manner.

1. First, we compute the geometrical mapping $\widetilde{\boldsymbol{\phi}}^{\mathrm{geo}}$ that maps $\Omega_0$ onto $\widetilde{\Omega}$, then we project $\widetilde{\boldsymbol{\phi}}^{\mathrm{geo}} - \boldsymbol{Id}$ on the POD basis $\{\boldsymbol{\zeta}_i^{\mathrm{geo}}\}_{1 \leq i \leq n}$ to obtain the coefficient $\widetilde{\alpha}$.

2. We use the pair $(\widetilde{\alpha}, \widetilde{\mu})$ to infer $\widetilde{\beta} = \mathcal{R}(\widetilde{\alpha}, \widetilde{\mu})$ and $\widetilde{\gamma} = \mathcal{O}(\widetilde{\alpha}, \widetilde{\mu})$. Then we have $\widetilde{\phi}^{\,\mathrm{opt}} :=$
$$\boldsymbol{Id} + \sum_{j=1}^{t} \widetilde{\beta}_j \boldsymbol{\zeta}_j^{\mathrm{opt}} \text{ and } \widetilde{u}^{\mathrm{opt}} := \sum_{j=1}^{r} \widetilde{\gamma}_j \psi_j.$$

3. Finally, the quantity of interest $\widetilde{u}$ is obtained as $\widetilde{u} := \widetilde{u}^{\mathrm{opt}} \circ \left(\widetilde{\phi}^{\,\mathrm{opt}}\right)^{-1} \circ \left(\widetilde{\phi}^{\,\mathrm{geo}}\right)^{-1}$.

# 4   Geometrical mapping

Numerous techniques exists in the literature to construct a mapping between domains [2, 4, 31]. In this work, we use RBF morphing [10] to construct the mapping from the reference domain $\Omega_0$ to each target domain $\Omega_i$. When using this technique, we suppose that the deformation on a subset of points in $\Omega_0$, called the control points, is known. These points are usually on the boundary of $\Omega_0$. Then, we can leverage the knowledge of $\phi_i(\partial\Omega_0) = \partial\Omega_i$ to compute the deformation in the bulk of the domain $\phi_i(\Omega_0)$. In this work, we suppose that the geometries are not parametrized and $\phi_i(\partial\Omega_0)$ is not given. Thus, we start by computing $\phi_i(\partial\Omega_0)$, for all $1 \leq i \leq n$. When computing $\phi_i(\partial\Omega_0)$ is not feasible, methods proposed in [11, 15] provides solutions to automatically finds a mapping from the reference onto the target domain.

# 5   Optimal mapping

The second major building block we introduce is the optimal mapping algorithm to compute $\Phi^{\mathrm{opt}} = (\boldsymbol{\phi}_i^{\mathrm{opt}})_{1 \leq i \leq n}$ that maximize the energy in the first $r$ modes.

In order to pass from the reference configuration to the optimal configuration, we solve (3):
$$\text{find } \Phi^* \in \arg\max_{\Phi \in \mathbf{M}_0^n} J_{\mathcal{U},r}[\Phi].$$

The optimization problem presented here is non-convex with possibly infinitely many solutions. We present here briefly the optimization strategy employed to solve this problem. Refer to the appendixes for a detailed discussion.

## 5.1   Gradient algorithm in infinite dimension

We can show that the differential of $J_{\mathcal{U},r}$ with respect to $\Phi$ at a point $\Psi$ has the following expression (see appendix A):
$$DJ_{\mathcal{U},r}[\Phi][\Psi] = \sum_{i=1}^{n} \int_{\Omega_0} \mathbf{f}_i[\Phi] \cdot \boldsymbol{\psi}_i dx.$$

with
$$\mathbf{f}_i[\Phi] := \sum_{j=1}^{n} \sum_{k=1}^{r} \left( \frac{2\zeta_{k,i}^{\mathcal{U}}[\Phi]\zeta_{k,j}^{\mathcal{U}}[\Phi]}{Tr(C_{\mathcal{U}}[\Phi])} - \frac{2\lambda_k^{\mathcal{U}}[\Phi]}{Tr(C_{\mathcal{U}}[\Phi])^2}\delta_{ij} \right) u_j^{\mathrm{ref}} \circ \phi_j \vec{\nabla} u_i^{\mathrm{ref}} \circ \phi_i. \tag{4}$$

A standard gradient algorithm in infinite dimension consists of (i) choosing an appropriate inner product, denoted here by $a$, (ii) computing the Riesz representation of each $\mathbf{f}_i[\Phi]$ with respect to $a$, which will be denoted by $\boldsymbol{u}_i[\Phi]$ and finally (iii) updating, at each iteration $m$, $\phi_i^{(m)}$ by
$$\phi_i^{(m+1)} = \phi_i^{(m)} + \epsilon \boldsymbol{u}_i[\Phi^{(m)}] \tag{5}$$

with $\epsilon > 0$ in the gradient step. The simplest example of $a$ is the $L_2(\Omega_0)$ inner product which gives the following iterative scheme:
$$\phi_i^{(m+1)} = \phi_i^{(m)} + \epsilon \mathbf{f}_i[\Phi^{(m)}]. \tag{6}$$

However, this may suffer from one of the following problems:

- $\phi_i^{(m+1)}$ might not be bijective.

- $\phi_i^{(m+1)}$ might not map $\Omega_0$ onto itself.

- If the function $\Phi^{(m)}$ is far from a maximum, the algorithm may fail to converge properly.

We address these issues in the following subsections.

## 5.2 Bijectivity

To address bijectivity, we consider the following optimization statement:

$$\text{find } \Phi^* \in \arg \max_{\Phi \in \mathbf{M}_0^n} I_{\mathcal{U},r}[\Phi] := J_{\mathcal{U},r}[\Phi] - c_1 \sum_{i=1}^n E[\phi_i], \tag{7}$$

where $E$ is an energy term to enforce bijectivity and $c_1 > 0$ is a penalization parameter. Ideally, the term $E$ should diverge to $+\infty$ if the mapping $\phi_i$ becomes non-bijective. In this case, the objective function $I$ diverges to $-\infty$. In this work, we use linear elastic energy for the term $E$, (we note that it does not diverges to $+\infty$ for non-bijective mappings, however, it gives acceptable results). Finally, we solve the problem 7 using the continuation method, which consists of iteratively decrease the parameter $c_1$ in order to obtain a solution to problem 3. We give more details about the continuation method and the term $E$ in appendix C.

## 5.3 Mapping condition

A necessary condition to map $\Omega_0$ onto itself is that the boundary of $\Omega_0$ is mapped also onto itself. If $\partial \Omega_0$ consists only of straight lines (faces in 3d), we impose that points on the boundary deform only in the tangential direction, so that they stay on the boundary. When $\partial \Omega_0$ presents also curves, we can fix points that lies on the curved part of $\partial \Omega_0$. Thus, we partition the boundary of $\Omega_0$ to $\partial \Omega_0 := \partial \Omega_0^p \bigcup \partial \Omega_0^c$ with $\partial \Omega_0^p$ is the union of all the straight lines (faces in 3d) of $\partial \Omega_0$ and $\partial \Omega_0^c$ is the curved part of $\partial \Omega_0$. We define the space $H_{\boldsymbol{n}}^1(\Omega_0) := \{\boldsymbol{u} \in H^1(\Omega_0) : \boldsymbol{u} \cdot \boldsymbol{n} = 0 \text{ on } \partial \Omega_0^p, \boldsymbol{u} = 0 \text{ on } \partial \Omega_0^c\}$

and the inner product $a$ on $H_{\boldsymbol{n}}^1(\Omega_0)$ as

$$(\boldsymbol{u}, \boldsymbol{v}) \mapsto a(\boldsymbol{u}, \boldsymbol{v}) := \int_{\Omega_0} \sigma(\boldsymbol{u}) : \varepsilon(\boldsymbol{v}) d\mathbf{x}, \tag{8}$$

with $\sigma$ and $\varepsilon$ are respectively the classical elasticity stress and strain tensors (we fix the Young modulus $E = 1$ and Poisson ratio $\mu = 0.3$ in this work). We compute, for all $1 \leq i \leq n$, and for each iteration $m$, the Riesz representation $\boldsymbol{u}_i^{(m)}$ of $\mathbf{f}_i$ with respect to this inner product, which is the unique solution to

$$\forall \boldsymbol{v} \in H_{\boldsymbol{n}}^1(\Omega_0; \mathbb{R}^d), \quad a(\boldsymbol{u}_i^{(m)}, \boldsymbol{v}) = \int_{\Omega_0} \mathbf{f}_i[\Phi^{(m)}] \boldsymbol{v} d\mathbf{x}, \tag{9}$$

to obtain the iterative scheme:

$$\phi_i^{(m+1)} = \phi_i^{(m)} + \epsilon \boldsymbol{u}_i[\Phi^{(m)}]. \tag{10}$$

We can also obtain a similar expression when solving problem (7) and using linear elastic energy (see more in appendix C):

$$\phi_i^{(m+1)} = \phi_i^{(m)} + \epsilon \left( \boldsymbol{u}_i[\Phi^{(m)}] - c_1 \left( \phi_i^{(m)} - Id \right) \right). \tag{11}$$

By using the above procedure, we guarantee that, if $\phi_i^{(m)} + \epsilon \boldsymbol{u}_i[\Phi^{(m)}]$ is bijective, then necessary we have that the points on $\partial \Omega_0^p$ stay on $\partial \Omega_0^p$, thus, preserving the boundary of $\Omega_0$. We note however that this is suboptimal as we do not allow $\partial \Omega_0^c$ to deform in this case. A full treatment of curved boundaries is the subject of future work.

## 5.4 Uncorrelated samples

One major reason for the poor convergence of the gradient algorithm is when the samples are heavily not correlated. This happens when the non-zero values of $u_i^{\text{ref}}$ are compactly supported in $\Omega_0$. More precisely, if, for some $1 \leq j \leq n$, $\text{supp}(u_j^{\text{ref}} \circ \phi_j) \bigcap \text{supp}(\vec{\nabla} u_i^{\text{ref}} \circ \phi_i) = \emptyset$, then the contribution of $u_j^{\text{ref}} \circ \phi_j \vec{\nabla} u_i^{\text{ref}} \circ \phi_i$ in $\mathbf{f}_i$ is null. To this end, we transform the fields $\{u_i\}_{1 \leq i \leq n}$ to the family of fields $\widehat{\mathcal{U}}(c_2) := \{\widehat{u}_i\}_{1 \leq i \leq n}$, where each is defined as the solution to

$$-\Delta \widehat{u}_i + c_2 \widehat{u}_i = c_2 u_i, \tag{12}$$
$$\partial_n \widehat{u}_i = 0$$

where $\Delta$ is the Laplacian operator and $c_2 > 0$. This equation has a unique solution. The transformed fields will diffuse the value of $u_i$ throughout the domain $\Omega_0$ for small values of $c_2$ (see Figures 3-4). On the other hand, $\widehat{u}_i$ converges to $u_i$ as $c_2$ goes to $+\infty$. We use the fields $\widehat{\mathcal{U}}(c_2)$ in order to solve (3)-(7). The value of $c_2$ is also changed throughout the iterations using the continuation method.

## 6 Numerical example

In this section, we illustrate the method on two example of non-reducible problems.

### 6.1 Example 1: advection-reaction equation

This example is taken from [30], and it illustrates a non-reducible problem. Here, we show the effect of the optimal mapping algorithm on a fixed geometry. Let the following advection-reaction equation:

$$\begin{cases} \nabla \cdot (c_\mu u_\mu) + \sigma_\mu u_\mu = f_\mu & \text{in } \Omega = (0,1)^2, \\ u_\mu = u_{D,\mu} & \text{on } \Gamma_{\text{in},\mu} := \{x \in \partial\Omega : c_\mu \cdot \boldsymbol{n} < 0\}, \end{cases}$$

where $\boldsymbol{n}$ denotes the outward normal to $\partial\Omega$, and

$$c_\mu = \begin{bmatrix} \cos(\mu_1) \\ \sin(\mu_1) \end{bmatrix}, \quad \sigma_\mu = 1 + \mu_2 e^{x_1+x_2}, \quad f_\mu = 1 + x_1 x_2,$$

$$u_{D,\mu} = 4 \arctan\left(\mu_3\left(x_2 - \frac{1}{2}\right)\right)(x_2 - x_2^2),$$

$$\mu = [\mu_1, \mu_2, \mu_3] \in \mathcal{P} := \left[-\frac{\pi}{10}, \frac{\pi}{10}\right] \times [0.3, 0.7] \times [60, 100].$$

We consider $n = 250$ samples. In Figure 2, we plot $u_\mu$ for three different values of the parameter before and after computing the optimal mappings that maximize $J_{\mathcal{U},r}$, for $r = 1$. The purpose of this first example is to demonstrate the effect of the optimal mapping algorithm on the fields. We can clearly see how they are aligned and can be approximated using a low dimensional space. In this scenario, the regression task becomes much easier. The optimal mappings algorithm can be

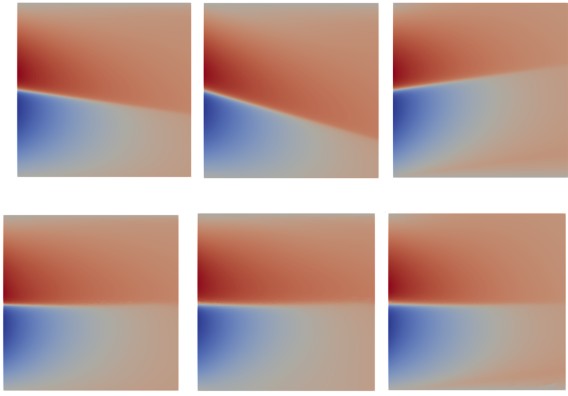

Figure 2: Top: three samples before the optimization. Bottom: three samples after the optimization.

seen as a multi-modal generalization to the registration methods, where the case $r = 1$ gives similar results to aligning all the samples on one mode. However, the advantage of our method is that we can go beyond a single mode. Furthermore, the alignment is automatic and does not any of the feature detection and tracking methods.

### 6.2 ML4CFD NeurIPS 2024 competition

In the second example, we apply the method to the airfoil design case considered in the ML4CFD NeurIPS 2024 competition [34], and we compare it with the winning solution. The dataset adopted for the competition is the AirfRANS dataset from [6]. Each sample have two scaler inputs, the inlet velocity and the angle of attack, and three output fields, the velocity, pressure and the turbulent viscosity. The dataset is composed of three splits.

1. Training set: composed of 103 samples.
2. Testing set: composed of 200 samples.

232  3. OOD testing set: composed of 496 samples, where the Reynold number considered for
233     samples in this split is taken out of distribution.

234  In this work, we focus on the turbulent viscosity field as it represents a non-reducible field. We
     illustrate this field in Figure 3.

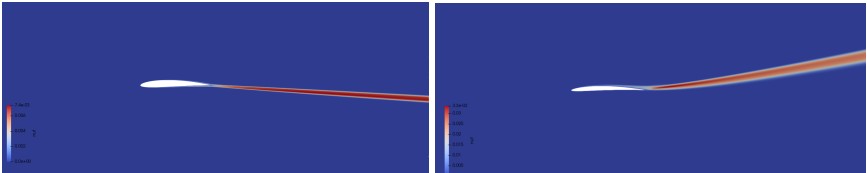

Figure 3: The turbulent viscosity field illustrated for two of the samples.

235

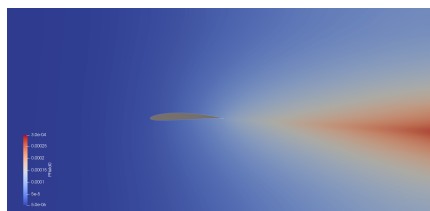

Figure 4: The diffused field $\widehat{u}$ for $c_2 = 1$ .

236  **6.3  NeurIPS solution: MMGP + wake line prediction**

237  The winning solution to the ML4CFD competition [7] relies on the application of the MMGP
238  method[8] , with the addition of aligning the wake line behind the airfoil for all the samples at same
239  position. The latter step being necessary to obtain accurate prediction of the turbulent viscosity.
240  While this correction step gives accurate results, it remains case-dependent and needs to be done
241  manually. Moreover, it suffers from the same problem as any other registration technique, which is
242  the restriction to one mode.

243  **6.4  O-MMGP: optimal mesh morphing Gaussian process**

244  The method we present in this paper is founded on the MMGP method [8]. The main differences
245  being:

246  1. First, in MMGP, the mapping $\phi_i^{\text{geo}}$ is computed from each geometry onto the reference
247     domain $\Omega_0$. In this case, the input to the GP model is the POD coefficients of the inverse
248     mapping $(\phi_i^{\text{geo}})^{-1}$. Thus, evaluating $(\phi_i^{\text{geo}})^{-1}$ would introduce extra computations to the
249     procedure. In addition, when using RBF morphing, such as in this work, computing $\phi_i^{\text{geo}}$
250     from the same geometry proves to be much efficient numerically, as the RBF interpolation
251     matrix $\mathbf{K}$ can be assembled and factorized once and for all (see appendix E for the definition
252     of $\mathbf{K}$).

253  2. Secondly, the MMGP mappings are not field-optimized. This is equivalent to omitting the
254     transition to the optimal configuration. So, when it comes to non-reducible problems, even
255     if a large number of modes are retained, the error introduced by POD truncation and the
256     discretization error would remain large to obtain reliable predictions for new samples.

257  We run the optimal mapping algorithm in order to maximize the compression of the turbulent
258  viscosity field. We choose the reference geometry $\Omega_0$ to be one of the samples in the training set. We
259  then proceed to compute $\phi_i^{\text{geo}}$ using RBF morphing. Once computed, we solve problem 7 for the
260  family $\widehat{\mathcal{U}}(c_2)$. We choose $r := 1$, $c_1^{(0)} := 0.1$ and $c_2^{(0)} := 1$. The last two parameters are changed
261  throughout the iterations as mentioned in appendix C. In Figure 5a, we show that the eigenvalues of
262  the correlation matrix of $(\phi_i^{\text{opt}})_{1 \le i \le n}$ decay rapidly, proving numerically that this family is reducible,

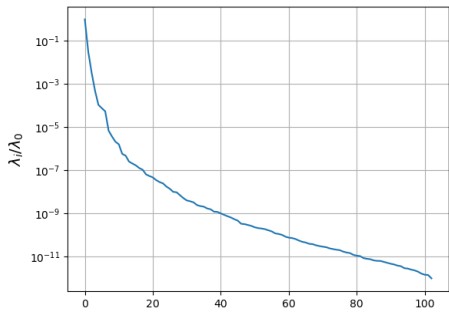
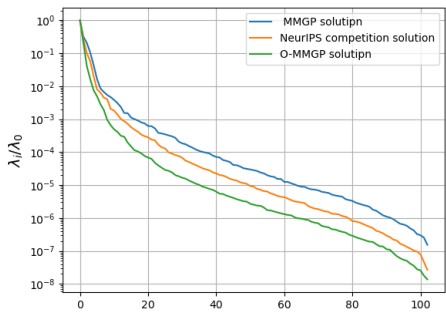

(a) Decay of the eigenvalues of the correlation matrix of the family $(\phi_i^{\mathrm{opt}})_{1 \le i \le n}$.

(b) Decay of the eigenvalues of the correlation matrix for the turbulent viscosity fields.

Figure 5: Decay of the eigenvalues.

and justifying the regression on the optimal mapping POD coordinates. The cost of the optimal mapping computation is discussed is appendix D

After computing the optimal mappings, we train two Gaussian process regression models, one to learn the optimal mapping $\widetilde{\phi}$, and the other to learn the field $\widetilde{u}$, which is the turbulent viscosity in this case.

In order to show the efficiency of the method, we compare the results of the following three tests:

1. First, we apply the method without solving the optimal mappings problem, similar to the original MMGP method. Thus, we predict the turbulent viscosity field in the reference configuration.

2. Second, we run the winning solution of the NeurIPS 2024 challenge, aligning the wake line behind the airfoil manually.

3. Finally, we apply the full O-MMGP procedure described in this paper.

In table 1, we report the mean square error (MSE) of the three tests on the turbulent viscosity field, for the testing and OOD splits. While the original MMGP performs poorly for this field, the O-MMGP method produces very accurate predictions, slightly surpassing the NeurIPS 2024 competition solution. The major advantage of the method is that the alignment of the snapshot was done automatically, without using the specificity of the case. In Figure 5b, we report the decay of the

|      | MMGP  | NeurIPS solution | O-MMGP |
|------|-------|------------------|--------|
| Test | 0.143 | 0.025            | 0.024  |
| OOD  | 0.171 | 0.048            | 0.046  |

Table 1: MSE errors for the different tests.

eigenvalues of the correlation matrix for the three tests. For the MMGP solution, no aligning of the solution was performed, which explains the slow decay of the eigenvalues for this case. As expected, the eigenvalues decays the most rapidly when using the optimal mappings algorithm.

# 7   Conclusion

In this article, we have presented a new algorithm that aims to maximize the energy in the first $r$ principal modes of a family of functions. The novelty of the algorithm lies in the fact that it automatically aligns the functions on an arbitrary number of modes, unlike other methods in the literature which may require feature tracking of the solution or assume that the family of functions can be compressed onto a single mode. We have shown how the proposed method can be integrated into the MMGP workflow to learn and predict the solutions of PDEs with geometric variabilities. The quality of the prediction was found to be on a par with state-of-the-art methods. Current work

aims to solve the problem in the general case of curved boundaries and provide a more thorough mathematical analysis of the method, as well as applying the method in the presence of multiple shocks.

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

# A Differential of $J_{\mathcal{H},r}$

Let $\Phi = (\phi_i)_{1 \le i \le n} \in \mathbf{M}$, $\Psi = (\psi_i)_{1 \le i \le n}$ a "small" variation around $\Phi$ and $\epsilon \in \mathbb{R}, \epsilon << 1$. To evaluate the differential of $J_{\mathcal{H},r}$ at a point $\Phi$, we evaluate $J_{\mathcal{H},r}$ at $\bar{\Phi} := \Phi + \epsilon\Psi$ and calculate:

$$DJ_{\mathcal{H},r}[\Phi][\Psi] = \lim_{\epsilon \to 0} \frac{J_{\mathcal{H},r}(\Phi + \epsilon\Psi) - J_{\mathcal{H},r}(\Phi)}{\epsilon} \tag{13}$$

We have:

$$\forall i, \ h_i \circ (\phi_i + \epsilon\psi_i)(x) = h_i(\phi_i(x) + \epsilon\psi_i(x))$$
$$\simeq h_i(\phi_i(x)) + \epsilon\vec{\nabla}h_i(\phi_i(x)) \cdot \psi_i(x)$$

where we neglect higher order terms. Next we evaluate :

$$C_{\mathcal{H},ij}[\Phi + \epsilon\Psi] = \langle h_i \circ (\phi_i + \epsilon\psi_i), h_j \circ (\phi_j + \epsilon\psi_j)\rangle_{L^2(\Omega_0)}$$
$$= \int_{\Omega_0} h_i \circ (\phi_i + \epsilon\psi_i)(x)h_j \circ (\phi_j + \epsilon\psi_j)(x)dx$$
$$= \int_{\Omega_0} h_i \circ \phi_i(x)h_j \circ \phi_j(x)dx + \epsilon\int_{\Omega_0} h_i \circ \phi_i(x)\vec{\nabla}h_j(\phi_j(x)) \cdot \psi_j(x)dx \tag{14}$$
$$+ \epsilon\int_{\Omega_0} \vec{\nabla}h_i(\phi_i(x)) \cdot \psi_i(x)h_j \circ \phi_j(x)dx + \epsilon^2\int_{\Omega_0} \vec{\nabla}h_i(\phi_i(x))\psi_i(x)\vec{\nabla}h_j(\phi_j(x))\psi_j(x)dx$$
$$:= C_{\mathcal{H},ij}[\Phi] + \epsilon DC_{\mathcal{H},ij}[\Phi][\Psi] + \epsilon^2 D^2 C_{\mathcal{H},ij}[\Phi][\Psi]$$

and

$$Tr(C_{\mathcal{H}}[\Phi + \epsilon\Psi]) = \sum C_{ii}[\Phi + \epsilon\Psi]$$
$$= Tr(C_{\mathcal{H}}[\Phi]) + 2\epsilon\sum_{i=1}^n \int_{\Omega_0} h_i \circ \phi_i(x)\vec{\nabla}h_i(\phi_i(x)) \cdot \psi_i(x)dx + \epsilon^2\sum_{i=1}^n \int_{\Omega_0} [\vec{\nabla}h_i(\phi_i(x)) \cdot \psi_i(x)]^2 dx$$
$$= Tr(C_{\mathcal{H}}[\Phi]) + \epsilon Tr(DC_{\mathcal{H}}[\Phi][\Psi]) + \epsilon^2 D^2 C_{\mathcal{H}}[\Phi][\Psi]$$

with $DC_{\mathcal{H},ij}[\Phi][\Psi] := \int_{\Omega_0} h_i \circ \phi_i(x)\vec{\nabla}h_j(\phi_j(x)) \cdot \psi_j(x)dx + \int_{\Omega_0} \vec{\nabla}h_i(\phi_i(x)) \cdot \psi_i(x)h_j \circ \phi_j(x)dx$
and $DC_{\mathcal{H}}[\Phi][\Psi] = (DC_{\mathcal{H},ij}[\Phi][\Psi])_{1 \le i,j \le n}$.

Next we evaluate $D\lambda_i^{\mathcal{H}}[\Phi][\Psi]$. We have $\forall i$ :

$$||\zeta_i^{\mathcal{H}}[\Phi]||^2 = 1$$

Now taking the differential on both sides, we get

$$\langle D\zeta_i^{\mathcal{H}}[\Phi][\Psi], \zeta_i^{\mathcal{H}}[\Phi]\rangle = 0$$

Using the fact that $C_{\mathcal{H}}[\Phi]\zeta_i^{\mathcal{H}}[\Phi] = \lambda_i^{\mathcal{H}}[\Phi]\zeta_i^{\mathcal{H}}[\Phi]$, we get again by taking the differential on both sides

$$DC_{\mathcal{H}}[\Phi][\Psi]\zeta_i^{\mathcal{H}}[\Phi] + C_{\mathcal{H}}[\Phi]D\zeta_i^{\mathcal{H}}[\Phi][\Psi] = D\lambda_i^{\mathcal{H}}[\Phi][\Psi]\zeta_i^{\mathcal{H}}[\Phi] + \lambda_i^{\mathcal{H}}[\Phi]D\zeta_i^{\mathcal{H}}[\Phi][\Psi]$$

We multiply the last equation by $\zeta_i^{\mathcal{H}}[\Phi]$ to get

$$(\zeta_i^{\mathcal{H}}[\Phi])^T DC_{\mathcal{H}}[\Phi][\Psi]\zeta_i^{\mathcal{H}}[\Phi] + 0 = D\lambda_i^{\mathcal{H}}[\Phi][\Psi] + 0$$

Thus we have finally

$$\lambda_i^{\mathcal{H}}[\Phi + \epsilon\Psi] = \lambda_i^{\mathcal{H}}[\Phi] + (\zeta_i^{\mathcal{H}}[\Phi])^T DC_{\mathcal{H}}[\Phi][\Psi]\zeta_i^{\mathcal{H}}[\Phi] \quad, \forall 1 \le i \le n \tag{15}$$

Taking the sum over the first $r$ eigenvalues, we obtain:

$$\sum_{j=1}^r \lambda_j^{\mathcal{H}}[\Phi + \epsilon\Psi] = \sum_{j=1}^r \lambda_i^{\mathcal{H}}[\Phi][\Psi] + \epsilon Tr((Z_r^{\Phi})^T DC_{\mathcal{H}}[\Phi][\Psi]Z_r^{\Phi}) \tag{16}$$

 with $Z_r^\Phi = (\zeta_1^\Phi, \zeta_2^\Phi, \cdots, \zeta_r^\Phi)^T$. Now we can evaluate (13) to obtain:

$$DJ_{\mathcal{H},r}[\Phi][\Psi] = \frac{Tr((Z_r^\Phi)^T DC_{\mathcal{H}}[\Phi][\Psi] Z_r^\Phi)}{Tr(C_{\mathcal{H}}[\Phi])} - \frac{\sum_{k=1}^{r} \lambda_k^{\mathcal{H}}[\Phi]}{Tr(C_{\mathcal{H}}[\Phi])^2} \times Tr(DC_{\mathcal{H}}[\Phi][\Psi]) \qquad (17)$$

732 which can be written explicitly as

$$DJ_{\mathcal{H},r}[\Phi][\Psi] = \frac{2}{Tr(C_{\mathcal{H}}[\Phi])} \sum_{i=1}^{n} \sum_{j=1}^{n} \sum_{k=1}^{r} \zeta_{k,i}^{\mathcal{H}}[\Phi] \zeta_{k,j}^{\mathcal{H}}[\Phi] \int_{\Omega_0} h_j \circ \phi_j(x) \vec{\nabla} h_i(\phi_i(x)) \cdot \psi_i(x) dx$$

$$- \frac{2 \sum_{k=1}^{r} \lambda_k^{\mathcal{H}}[\Phi]}{Tr(C_{\mathcal{H}}[\Phi])^2} \sum_{i=1}^{n} \int_{\Omega_0} h_j \circ \phi_j(x) \vec{\nabla} h_i(\phi_i(x)) \cdot \psi_i(x) dx$$

$$= \sum_{i=1}^{n} DJ_i[\Phi][\psi_i]$$

733 with

$$DJ_i[\Phi][\psi_i] := \sum_{k=1}^{r} \left( \frac{2\zeta_{k,i}^{\mathcal{H}}[\Phi]^2}{Tr(C_{\mathcal{H}}[\Phi])} - \frac{2\lambda_k^{\mathcal{H}}[\Phi]}{Tr(C_{\mathcal{H}}[\Phi])^2} \right) \int_{\Omega_0} h_i \circ \phi_i(x) \vec{\nabla} h_i(\phi_i(x)) \cdot \psi_i(x) dx$$

$$+ \sum_{\substack{j=1 \\ j \neq i}}^{n} \sum_{k=1}^{r} \frac{2\zeta_{k,i}^{\mathcal{H}}[\Phi] \zeta_{k,j}^{\mathcal{H}}[\Phi]}{Tr(C_{\mathcal{H}}[\Phi])} \int_{\Omega_0} h_j \circ \phi_j(x) \vec{\nabla} h_i(\phi_i(x)) \cdot \psi_i(x) dx$$

$$= \sum_{j=1}^{n} \sum_{k=1}^{r} \left( \frac{2\zeta_{k,i}^{\mathcal{H}}[\Phi] \zeta_{k,j}^{\mathcal{H}}[\Phi]}{Tr(C_{\mathcal{H}}[\Phi])} - \frac{2\lambda_k^{\mathcal{H}}[\Phi]}{Tr(C_{\mathcal{H}}[\Phi])^2} \delta_{ij} \right) \int_{\Omega_0} h_j \circ \phi_j(x) \vec{\nabla} h_i(\phi_i(x)) \cdot \psi_i(x) dx,$$

$$(18)$$

734 and $\delta_{ij}$ is the Kronecker delta. For simplicity, we note the term

$$\mathbf{f}_i[\Phi] := \sum_{j=1}^{n} \sum_{k=1}^{r} \left( \frac{2\zeta_{k,i}^{\mathcal{H}}[\Phi] \zeta_{k,j}^{\mathcal{H}}[\Phi]}{Tr(C_{\mathcal{H}}[\Phi])} - \frac{2\lambda_k^{\mathcal{H}}[\Phi]}{Tr(C_{\mathcal{H}}[\Phi])^2} \delta_{ij} \right) h_j \circ \phi_j(x) \vec{\nabla} h_i(\phi_i(x)), \qquad (19)$$

$\mathbf{F}[\Phi] := (\mathbf{f}_1[\Phi], \cdots, \mathbf{f}_n[\Phi])$, and we write

$$DJ_{\mathcal{H},r}[\Phi][\Psi] = \sum_{i=1}^{n} \int_{\Omega_0} \mathbf{f}_i[\Phi] \cdot \psi_i dx.$$

## B  Differential of $I_{\mathcal{H},r}$

We define the objective function $I_{\mathcal{H},r}$ on $M_0^n$ as

$$I_{\mathcal{H},r}[\Phi] := J_{\mathcal{H},r}[\Phi] - c_1 \sum_{i=1}^{n} E[\phi_i].$$

As mentioned in section 5, the energy term used in this work in the linear elastic energy defined as

$$E[\phi_i] := \frac{1}{2} a(\phi_i - \mathbf{Id}, \phi_i - \mathbf{Id}),$$

where we recall the definition of $a$ in (8). We can easily show that

$$DE[\phi_i][\psi] = a(\phi_i - \mathbf{Id}, \psi).$$

Using the linearity of the differential, we obtain the differential of $I_{\mathcal{H},r}$ as

$$DI_{\mathcal{H},r}[\Phi][\Psi] = DJ[\Phi][\Psi] - c_1 \sum_{i=1}^{n} DE[\phi_i][\psi_i]$$

$$= \sum_{i=1}^{n} \int_{\Omega_0} \mathbf{f}_i[\Phi] \cdot \psi_i dx - c_1 \sum_{i=1}^{n} DE[\phi_i][\psi_i]$$

$$= \sum_{i=1}^{n} \left( \int_{\Omega_0} \mathbf{f}_i[\Phi] \cdot \psi_i dx - c_1 DE[\phi_i][\psi_i] \right)$$

$$:= \sum_{i=1}^{n} DI_i[\Phi][\psi_i].$$

In order to obtain equation (11), we compute the Riesz representation of $DI_i[\Phi]$, denoted as $\bar{\boldsymbol{u}}$, with respect to the inner product $a$, the unique solution to

$$a(\bar{\boldsymbol{u}}_i, \boldsymbol{\psi}) = \int_{\Omega_0} \mathbf{f}_i[\Phi] \cdot \psi_i dx - c_1 a(\boldsymbol{\phi}_i - \boldsymbol{Id}, \boldsymbol{\psi}),$$

for all test function $\boldsymbol{\psi}$. Since we also have $\boldsymbol{u}_i$ as the Riesz representation of $DJ_i[\Phi]$ $(a(\boldsymbol{u}_i, \boldsymbol{\psi}) = DJ_i[\Phi][\boldsymbol{\psi}])$, we obtain

$$a(\bar{\boldsymbol{u}}_i, \boldsymbol{\psi}) = a(\boldsymbol{u}_i, \boldsymbol{\psi}) - c_1 a(\boldsymbol{\phi}_i - \boldsymbol{Id}, \boldsymbol{\psi}).$$

which is true for all test function $\boldsymbol{\psi}$. Thus, by the linearity of $a$, we obtain $\bar{\boldsymbol{u}}_i = \boldsymbol{u}_i - c_1(\boldsymbol{\phi}_i - \boldsymbol{Id})$, and hence equation (11).

# C  Continuation method

We give here a quick description on the continuation methods. Readers can refer to [1, 27] and references within for more details.

Continuation methods are a variety of methods used to solve equations of the type

$$G(x) = 0$$

where $G$ is a non-linear function, and $x$ is the unknown. Continuation methods rely on solving a succession of problems of the form

$$H(x, \lambda) = 0$$

where $H$ is another non-linear function and $H(x, \lambda^*) = G(x)$ for some $\lambda^*$. $\lambda$ is called the continuation parameter. In the simplest form of a continuation method, we start by choosing a set a values $\{\lambda^0, \lambda^1 \cdots \lambda^*\}$ for $\lambda$. We then solve $H(x^i, \lambda^i) = 0$, and initialize $H(x^{i+1}, \lambda^{i+1}) = 0$ with $x^i$.

## C.1  Continuation for $c_1$

In order to solve equation (7), we proceed by using the continuation method as follows:

1. We start by solving equation the optimization problem, using the gradient algorithm, for $c_1 = c_1^0$, where $c_1^0$ should be sufficiently large in order to converge properly. This will produce the function $\Phi^0$.

2. At each iteration $k$, we set $c_1^k := \dfrac{c_1^{k-1}}{2}$, and we resolve the problem for $\Phi^k$ by initializing $\Phi^{(0),k} = \Phi^{k-1}$.

3. We iterate over $k$ as long as there are no elements of the morphed meshes are inverted.

We give a few comments about the above procedure. First, we note that there are two loops for the algorithm: the outer loop over $k$ that updates $c_1$, and the inner loop that solves the non-linear equation using the gradient algorithm. Second, ideally we want to solve (7) for $c_1 = 0$. If, while iterating over the outer loop, $c_1^k$ gets sufficiently small, we can set $c_1^{k+1} = 0$. Finally, we note that full convergence for intermediate results of $\Phi^k$ is generally not necessary.

## C.2  Continuation for $c_2$

Applying continuation with two or more parameters proves to be more complicated for the case of one parameter. To this end, we propose in this paper to alternate changing the values of the two parameters $c_1$ and $c_2$. We proceed in the same manner as above, expect that:

1. At iteration $2k$, we set $c_1^{2k} := \frac{c_1^{2k-1}}{2}$.

2. We start with a small value for $c_2^0$. At iteration $2k+1$, we set $c_2^{2k+1} := 10 \times c_2^{2k}$.

## C.3  Initialization and scheduling strategy

In our numerical examples, we find that the method is not sensitive for the initial choices for both parameters as expected. In fact, choosing a large value for $c_1$ would force the mapping to be close to the identity map. In this case, the mesh would not deform by much and the algorithm would converge quickly, and in return we change the value of $c_1$. On the other hand, choosing a small value of $c_2$ would have a similar effect in terms of convergence. A small value would diffuse very much the values of the field in the domain, and they approach a constant field allowing to converge rapidly and to change the values of $c_2$ in return.

The scheduling strategy used to change to values of $c_1$ and $c_2$ is a simple heuristic that we observed to work well with the tests provided. Other strategies would be to change both values at the same time, which also worked well for the same datasets. However, we do mention that other continuation methods exists when dealing with more than one parameter as in the references provide. These methods are developed for non linear problems in finite dimension and could be potentially harder to implement in functional settings as in this work.

# D  Computational cost of the optimal mapping

The training phase of O-MMGP consists of i) computing $\phi_i^{geo}$, ii) computing $\phi_i^{opt}$, iii) performing POD and iv) training of the gaussian processes to learn $\phi^{opt}$ and $u^{opt}$. With respect to MMGP, this would add step iii, and one GP training (for $\phi^{opt}$). For the AirfRANS dataset, steps i, iii, and iv take about 40 seconds. Step ii consists also of multiples steps (remeshing the reference mesh to tackle computational bottlenecks (see below), the optimization process, computing $\widetilde{u}(\alpha)$ and $\nabla\widetilde{u}(\alpha)$ ...). It takes about 35 minutes (we do mention however that some of these steps could be implemented in parallel which would greatly decrease the optimal mappings computational time).

Inference time: the inference takes about 81 seconds for 696 samples (200 test and 496 OOD). Both MMGP and NeurIPS 2024 solutions are faster to train (no optimization problem in offline), however they are both more expensive than O-MMGP in online. All the simulations are performed using 128 CPU cores and no GPU. In comparison, the full order solver takes about 25 min for each full simulation.

**Acceleration strategies**

Solving the optimal mappings optimization problem requires evaluating $F_i[\Phi]$ (thus $u_i$ and its gradient) at the integration points of the (deformed) reference mesh, the assembly of the stiffness matrix (once and for all for all the iterations), and solving the linear systems for the Riesz representation. Potential computational bottlenecks come from having a very fine reference mesh, which increases the degrees of freedom and the number of integration points. In this paper, we used an acceleration strategy consisting of using a coarse reference mesh to compute the optimal mappings, then the mappings are evaluated on the fine mesh using interpolation. This greatly decrease optimization time while giving accurate results. To further accelerate computation, when needed, other techniques can be used like reduced quadrature formulae or randomized linear algebra. However, this was not needed for the provided examples.

 # E   RBF morphing

801 In radial basis function (RBF) interpolation, we aim to approximate a scalar-valued function $g(x)$
802 using a linear combination of radial basis functions as follows:

$$g(x) \approx \sum_{i=1}^{n_c} a_i\, \xi(\|x - x_i\|), \tag{20}$$

803 where $a_i \in \mathbb{R}$ and $\xi$ is a radial basis function that depends only on the Euclidean distance between $x$
804 and a control point $x_i$. The control points $\{x_i \mid i = 1, \ldots, n_c\}$ are specific locations where the exact
805 values of $g(x)$, denoted $g_i$, are known. To determine the coefficients $a_i$, we enforce the interpolation
806 condition at all control points, yielding the system:

$$\sum_{i=1}^{n_c} a_i\, \xi(\|x_j - x_i\|) = g_j, \quad j = 1, \ldots, n_c. \tag{21}$$

807 This can be expressed in matrix form as:

$$\mathbf{K}\boldsymbol{a} = \boldsymbol{g}, \tag{22}$$

808 where $\mathbf{K}$ is an $n_c \times n_c$ interpolation matrix with entries $K_{ij} = \xi(\|x_j - x_i\|)$, $\boldsymbol{a} = [a_1, \ldots, a_{n_c}]^T$ is
809 the vector of coefficients, and $\boldsymbol{g} = [g_1, \ldots, g_{n_c}]^T$ is the vector of known values. Solving this linear
810 system yields the coefficients $a_i$, which can then be used to compute $g(x)$ at any point $x$.

811 In the context of radial basis function-based mesh morphing, RBF interpolation is applied separately
812 to each component of the morphing function $\boldsymbol{\phi}(x) = (\phi_1(x), \phi_2(x), \phi_3(x))$ in 3D (or similarly in
813 2D). The control points are usually chosen as boundary points. The interpolated morphing function is
814 then used to smoothly deform the interior of the mesh according to the control point displacements.

