# OpenReview forum: "O-MMGP: Optimal Mesh Morphing Gaussian Process Regression for Solving PDEs with non-Parametric Geometric Variations"
_NeurIPS.cc/2025/Conference — Submitted to NeurIPS 2025_

### Official Review · Reviewer_hWiP · 2025-06-24

**Clarity:** 2
**Significance:** 3
**Originality:** 3
**Rating:** 4
**Confidence:** 1

**Summary:**

The authors used an optimization process on Gaussian Processes, used as surrogates for the true functions that map parameters and "PODs" to "optimal PODs", to find the optimal configuration (parameters) for a complex physical problem, provided in the form of a PDE.

**Questions:**

- What is the difference between $\phi^\text{geo}$ and $\phi^\text{opt}$?
- Why don't you combine the GPs $\mathcal{R}$ and $\mathcal{O}$ into a single multi-output GP that learns $\gamma$ and $\beta$ together?
- Could you give an estimate on how hard/easy it is to find the mappings $\phi$ for various different problems? i.e. how easily applicable is your approach to different problems. Can it be done algorithmically or are all preparatory calculations done manually?

**Ethical Concerns:**

["NO or VERY MINOR ethics concerns only"]

**Final Justification:**

Since my points were minor and due to understanding little of the topic in general, even despite the authors explaining them I feel my score is appropriate and should not be raised.

**Limitations:**

- As mentioned in the questions, the difficulty of applying the authors method was not discussed

**Paper Formatting Concerns:**

The paper exceeds the nine page limit.

**Quality:**

3

**Strengths And Weaknesses:**

# Strengths
- The problem seems highly relevant and the example use case is well chosen
- While the required preliminaries are very advanced, the authors did ok in explaining the material in an understandable fashion. Still, there is room to provide intuitions and refer to the application areas throughout the discussion, making it even more understandable.

# Weaknesses
- **The authors did not keep to the nine pages limit. Gaining a (slight) unfair advantage over the remaining submissions.**

- Figure 1 is unclear, I can't see the second airfoil without zooming in to 200%

## Minor
- Positioning of the plots in Figure 2
### Typos
- 14: varibale
- 117: depends
- 148: provides, finds
- 152: maximize
- 182: lies
- 250: "proves to be much efficient numerically"
- Legend in Figure 5 b)
- 282: decays

---

> ### Author Rebuttal · Authors · 2025-07-29
>
> We thank the reviewer for their comments. We will make sure to correct the typos.
>
> We will more details to explain some key concepts like the Kolmogorov width and non reducible problems in the final version, with references to these key concepts.
>
>
> Weaknesses
>
> 1- All our excuses for passing the 9 page limit by few lines. We did not pay attention to that during submissions.
>
> 2- We will make sure to change the color of the airfoil for the final version.
>
> Questions:
>
> 1- $\phi^\text{geo}$ is the geometrical mapping. Given two geometries $\Omega_0$ and $\Omega_1$, $\phi^\text{geo}$ maps $\Omega_0$ onto $\Omega_1$. It is purely geometrical and does not depends on the physical problem or any other sample in the dataset.
>
> $\phi^\text{opt}$: it is the optimal mapping, solution to the proposed optimization problem. Solving for $\phi^\text{opt}$ is the main contribution in this paper. Given a reference geometry $\Omega_0$ and a dataset of geometries $\{\Omega_i\}_{1\leq i \leq n}$, each with a field $u_i$ defined on $\Omega_i$, $\phi_i^\text{opt}$ would align the fields $u_i\circ \phi^\text{geo}$ on the reference geometry so that they can be captured in a small number of modes. The computation of $\phi_i^\text{opt}$ for a sample $i$ depends on all the fields, so it is a coupled problem between all the samples. $\phi^\text{geo}$ can be viewed in this context as the initialization to compute $\phi^\text{opt}$.
>
> When there is no geometrical variation in the problem, that is all the geometries are all the same, we take  $\phi^\text{geo}=Id$, the identity map.
>
> 2- Indeed this is possible, we separate the two because the two models are conceptually different: one is to learn the mappings, and the other is to learn the fields.
>
> 3- To compute $\phi^\text{geo}$, this would depend on the problem. While some methods should be done manually, other try to extend these computation so that they can be done automatically. See section 4 for a brief discussion about these computations.
>
> The computation of  $\phi^\text{opt}$ is done automatically without any manual precomputation. This is a huge interest of the method with respect to other existing methods in the literature.

---

> > ### Comment · Reviewer_hWiP · 2025-08-06
> >
> > Thank you for clarifying certain points and showing me some of my misunderstandings or things I missed while reading.
> >
> > Since my points were minor and due to understanding little of the topic in general, even despite the authors explanations I feel my score is appropriate.
> >
> > I wish the authors the best with their contribution.

---

### Official Review · Reviewer_nmYu · 2025-06-28

**Clarity:** 3
**Significance:** 2
**Originality:** 2
**Rating:** 2
**Confidence:** 4

**Summary:**

This paper presents O-MMGP, a method for solving PDEs where the solution domains vary non-parametrically. The approach first morphs all domains to a common reference using RBF interpolation. Then, a mapping function is optimized to align solution fields such that energy is maximized in the first few POD modes. Once the alignment is done, POD is performed and two GPR models are trained: one to predict the mapping coefficients, and another to predict the reduced solution coefficients. At inference time, the method applies the predicted mappings and reconstructs the full solution. The method is tested on a synthetic advection-reaction example and on the AirfRANS dataset from ML4CFD, where it shows improved MSE over both MMGP and a manually aligned baseline.

**Questions:**

Just to note before the questions—some of the earlier concerns feel pretty central. I hope the authors take a moment to address those too.
- Can you explain how runtime and memory usage scale with bigger meshes, higher dimensions, or more training samples?
- Did you track how often the mapping step actually produces valid, bijective results? Were there any cases where the mapping folded or failed during training?
- Have you tried adding noise to the control points or tweaking the boundary shapes a bit? How much does that affect the mapping and final predictions?
- How sensitive is the method to things like the number of POD modes or the penalty and diffusion terms in the optimization? Any tips or heuristics for picking those values?
- The paper doesn’t test training on a coarse mesh and then running on a finer mesh to see if accuracy holds. Adding a simple “train coarse, test fine” experiment or discussing how interpolation might affect results would make the approach more trustworthy.
- Have you tested this approach on other types of PDEs or in 3D settings? If not, what kind of issues do you expect to run into?
- Why didn’t you compare to operator-learning methods like Geo-FNO or DeepONet applied to the aligned fields? Do you think those models might outperform your GPR approach?
- Does the uncertainty from GPR actually match the prediction error? For example, do variances go up when you move away from the training data?

**Ethical Concerns:**

["NO or VERY MINOR ethics concerns only"]

**Final Justification:**

After considering the rebuttal, I think the scope is limited and I maintain my rating.

**Quality:**

2

**Strengths And Weaknesses:**

### **Strengths**
- The method improves the alignment of solution fields, helping make low-rank approximation more effective—this is supported by the eigenvalue plots. It keeps the original MMGP structure intact, adding just one step offline while keeping online inference efficient.
- The optimization for alignment is explained clearly, with enough detail to follow. Using a coarse mesh for alignment and interpolating to a fine mesh is a simple but useful way to reduce cost.
- The geometric alignment could possibly transfer beyond the intended application, but this was not demonstrated.


### **Weaknesses**
- The scope is too limited: it is only relevant to those who are dealing with parametric PDEs with geometric variations AND engaging POD AND with the objective to compress ensemble from multiple geometry for training a surrogate like GPR. The framework cannot be applied to new end-to-end AI methods. It seems to me a niche application, not the groundbreaking work expected for NIPS papers.
- Do the PODs change based on initial conditions? How do we handle this? It seems the POD approach runs into many different challenges.
- The alignment step is based on a non-convex optimization over functions, but the paper doesn't provide any analysis of how reliably this process works. There’s no mention of how often the mappings remain valid (bijective) or whether they break down—like folding or other distortions—in practice.
- The method assumes that all domains have the same topology and that boundary control points can be matched across samples. This is a big ask in real-world problems, where clean correspondences are rare. - The paper doesn’t test or discuss how the method holds up when these assumptions don’t quite hold—like when control points are noisy or misaligned.
- There are several hyperparameters that clearly matter (number of modes, penalty weights, step sizes, etc.), but no ablation or sensitivity study is included. This makes it hard for others to know how to tune the method or trust that it will work out-of-the-box on different problems.
- The experiments are narrow: one toy problem and one 2D real-world dataset. There’s no test on 3D problems, other types of PDEs, or geometries with more extreme variation. Without that, it's hard to know if the method really works beyond the settings chosen here.
- The comparisons are also limited. The paper only compares against MMGP and a manually aligned baseline. There’s no comparison to modern operator-learning models like DeepONet or Geo-FNO, which could potentially be paired with the same alignment step. Without that, the improvement over existing methods feels isolated and hard to evaluate.
- Even though the method uses GPR, which provides predictive uncertainty, the paper doesn’t actually report or analyze any of it. There’s no look at calibration, error bars, or how uncertainty behaves in test or OOD settings. That’s a missed opportunity for a UQ-related contribution.
- Runtime is only given for a single case, and there’s no discussion of how cost scales with mesh size, number of samples, or problem dimension. That makes it difficult to understand how practical the method would be in larger or more realistic settings.
- The results are shown without any variability analysis—just single-run metrics with no error bars. It’s unclear how consistent or significant the improvements are, especially when the gains are small. This weakens confidence in the overall empirical contribution.

---

> ### Author Rebuttal · Authors · 2025-07-30
>
> We thank the reviewer for all the valuable remarks.
>
> Weaknesses
>
> 1- The scope of this work is to accelerate the simulation of parametric PDEs defined on variable geometries. The main novelty of the paper, which is the optimal mappings, can be viewed as a pretreatment step applied to the dataset. We  chose an approach based on non intrusive reduced order modeling  coupled with a learning algorithm. Other choices for the methodology and the learning algorithm can be applied to the transformed fields.  For example, we could couple the optimal mapping algorithm to transform the field to the optimal configuration, then apply other techniques like FNO on the transformed fields. This is indeed a work in progress. Internal tests illustrated that morphing pretreatment enabled 30% RMSE improvements of vision transformer-based methods. We chose the MMGP context as it falls exactly in the same context: instead of finding a mapping, we find an optimal mapping.  A limitation of our approach is the fixed topology, as we are seeking to compute bijective mappings.
>
> 2- Folding: this is a very valid point. Folding may occur if the energy penalization term is small or if the gradient step is large. In practice, we start with a large coefficient for the energy penalization term and decrease it using the continuation method.
>
> 3-  Topology and control points: indeed we suppose that the topology is constant across the dataset. This is a limitation of the approach that we have not yet addressed. This is due to the fact that we seek bijective mappings.
> The control points are not required to be matched across samples. In fact, we only have control points on the reference domain to apply the geometrical mapping using RBF morphing onto the target domains. We do not have any supposition of the boundary of the target domain. We match the boundaries by placing the nodes of the boundary of the reference domain on the boundary of the target domain following the curvilinear abscissa.
>
> 4- We disused the sensitivity of the initial choices of $c_1$ and $c_2$ in appendix C.3, repeated here for reference:
>  "In our numerical examples, we find that the method is not sensitive for the initial choices for both parameters as expected. In fact, choosing a large value for $c_1$ would force the mapping to be close to the identity map. In this case, the mesh would not deform by much and the algorithm would converge quickly, and in return we change the value of $c_1$. On the other hand, choosing a small value of $c_2$  would have a similar effect in terms of convergence. A small value would diffuse very much the values of the field in the domain, and they approach a constant field allowing to converge rapidly and to change the values of $c_2$ in return."
>
> Number of modes $r$ : the most delicate parameter between the three. If we do not have any intuition about the shape of the solutions, we should try different number of modes. Less modes is better for learning. Bigger number of modes: less mesh deformation thus decrease the risk of having non bijective mappings, and smaller optimization time (in the extreme case of $r=n$, the energy is maximal, so the identity is an optimal solution, the reference mesh does not deform and no optimization is required), but the prediction becomes much less accurate for non reducible problems. For both cases considered in the numerical tests, we choose $r=1$. We could also report the solution of the mappings for $r>1$ if necessary, as we consider this is an advantage in our method with other similar registration methods in the literature.
>
> Gradient step : fixed for all iterations in our experiments.
>
> 5- Experiments: the method is intended to be applied to non reducible problems. We have tested on a dataset with parabolic PDE, however the gain was minimal since the problem is already reducible. Here, we chose an advection reaction and the turbulent viscosity of the flow around naca profiles as they fall excatly in the intended context. A 3d test case is a perspective of the work. While the geomtric variation in not directly the main contribution of this work, we have shown how the compression method using the morphig can be integrated in this context. If we have more extreme geomtrical variations, this could render the problem of computing $\phi^{geo}$ more difficult. We refer to section 4 for more details.
>
> 6- Using other methodologies: Indeed we could pair the mapping algorithm with other technique than MMGP as a preprocessing step. This is a work in progress to verify if the algorithm facilitate other state of the art methods, not only with its intended scope (linear dimensionality reduction + regression algorithm on the reduced space)
>
> 7-  Given that uncertainty quantification was already systematically evaluated in [1], the present paper deliberately focuses on a complementary aspect: the optimal and automated construction of morphing maps. Our goal is to advance MMGP's practical deployment by addressing efficiency, generality, and usability, while considering the predictive uncertainty aspects as already well established in prior work. We will make this scope more explicit in the revision and direct readers to the original paper for details on predictive uncertainty.
>
> 8-  Cost: the cost is mainly determined by the size of the reference mesh and the number of samples. To mitigate the problem related to the reference mesh, we chose a coarse reference mesh to calculate the mappings, then evaluated the mappings on a finer mesh after solving the optimization problem. We did not try any strategies to solve the problem of the high number of samples, but grouping similar samples and using a reduced-order basis based on a dictionary could mitigate this problem by solving several smaller optimization problems.
>
> Response to questions
>
> 1- Scaling: see remark 7 above.
>
> 2- Bijectivity: we could track the bijectivity at each step. In practice, this would add cost to the optimization offline. We did check it every 10 iterations, although this was a heuristic. If the step size is large or if we start with a small value for the penalization term, the optimization could fail to converge.
>
> 3- Boundary points: the boundary control points are placed on the boundary of the target domain following the curvilinear abscissa. The quality of the alignment of the boundary points could greatly affect the results in the case of naca profiles, as the mesh near the profile is very fine, so small misalignment could introduce large errors when performing interpolation.
>
>  4- Sensitivity: see remark 4 above and appendix C
>
> 5- Train coarse, test fine: Training on a coarse mesh risks having the prediction error dominated by information loss during the interpolation step, as it will introduce interpolation error. The accuracy of the predictions would be capped by these errors. Since the cost is in computing $\phi^{opt}$, we compute them on the coarse mesh. Once done, we evaluate the rest of the finner mesh.
>
>
>
> 6- 3D settings: We have not tested untill now on a 3d test case, but this is a perspective of the work. While the method is formulated without taking into consideration the dimension of the problem, we could face some computational bottlenecks in the 3d test case.
>
> 7-  Operator-learning methods: testing other regression based methods on the transformed fields is a future work. At this time, we presented the optimal mapping algorithm and we showed its effect in its intended context, POD to reduce the dimension of the solution manifold + a regression algorithm on the reduced space.
>
> 8- Uncertainty: The predictive uncertainty provided by the variance of the Gaussian Processes (GPs) has been thoroughly addressed in the original MMGP paper [1]. In particular:
>
> - Table 2 reports standard deviations for both relative RMSE and Q² across all experiments.
> - Figure 4 illustrates the predicted output variance for the pressure field: the uncertainty is higher in regions where the error is also larger.
> - Figure 5 shows 95% confidence intervals with respect to a scalar input parameter; the intervals widen appropriately as the input moves away from the training range, demonstrating discriminative uncertainty quantification.
> - The Prediction Interval Coverage Probability (PICP) at the 95% level was computed as 93.05% and 93.5% for two scalar outputs in the AirfRANS experiment, confirming the reliability of the predictive intervals.
> - Figures 15 to 20 provide additional qualitative assessments: they show predicted values, GP variances, and errors. The variance is well correlated with the observed error; the reference solution lies within the confidence intervals for in-distribution test samples, while this property degrades—as expected—on out-of-distribution (OOD) data. The confidence intervals are also well calibrated: tight on the training set, wider on in-distribution test data, and substantially wider OOD.
>
>
> [1] Casenave, Fabien, Brian Staber, and Xavier Roynard. MMGP: a mesh morphing Gaussian process-based machine learning method for regression of physical problems under nonparametrized geometrical variability. Advances in Neural Information Processing Systems 36 (2023): 43972–43999.

---

> > ### Comment · Reviewer_nmYu · 2025-08-04
> >
> > While future work could integrate the mapping step with operator learners like FNO or DeepONet, no empirical support is provided. Could the author provide a toy problem at least?
> >
> > The method is validated only on one toy example and the AirfRANS 2D benchmark for a single variable (turbulent viscosity). Let N>1. No test on 3D, more complex boundary topologies, or realistic multi-physics PDEs, or even other problems. Given the morphing is sensitive to geometry and topology, this is a significant gap.
> >
> > No comparison to modern operator-learning methods (e.g., DeepONet, Geo-FNO), which the authors admit are compatible with the morphing. I think the work seems not finished yet.
> >
> > How do we know if bijectivity is robust? there is also no ablation on what happens when it fails.

---

> > > ### Author Response · Authors · 2025-08-06
> > >
> > > The geometrical and optimal mappings $\phi^{geo}$ and $\phi^{opt}$ can be applied as preproccessing steps to other methods in the same fashion to MMGP. In this settings, we learn the mapped field $u_i^{opt}$ (introduced in section 2) instead of learning directly the fields $u_i$. Up untill now, we coupled the application of $\phi^{geo}$ with vision based transformers to learn the output fields. We applied the mappings with vision transformers on a 2D quasistatic non-linear structural mechanics dataset. It consists of 500 train samples and 200 test samples. The inputs consists of 6 scalars and the mesh of the geometry, variable for each sample. The outputs are the stress and strain fields, and three scalar quantity of interest. Applying the mappings allowed to deacrease the RMSE on the total error of all the outputs by 50% in our latest experiments.
> > >
> > > In the same fashion, our preprocessing steps could also be coupled with FNO for example to learn $u_i^{opt}$. We will test coupling the two methods same as we did with the vision transformers above.
> > >
> > > Bijectivity: the continutation method we presented in the appendix allows us to remain always bijective. We start from a "rigid" mesh using a strong elsaticity penalization, and we relax the rigidity as we solve our optimization problem. Once we are no longer bijective, we assume that we hit our limit and we stop the iterations.
> > >
> > > Complex geometries: at this stage, the method perform well with polytope domains. In order to perfom well with curved boundaries, we should derive the conditions to stay in the admissible space. This is a work in progress. A naive solution would be to fix the nodes on the boundaries and only deform the interior. However, this would be sub-optimal and the gain would depend in the problem. For example, if we have a moving shock on the boundary of an airfoil, fixing the boundary nodes will not allows us to track the shock well.

---

### Official Review · Reviewer_NSkh · 2025-06-30

**Clarity:** 1
**Significance:** 2
**Originality:** 2
**Rating:** 3
**Confidence:** 1

**Summary:**

The author proposes an optimal mesh morphing Gaussian process (O-MMGP) to achieve the non-linear dimensionality reduction for efficient and accurate modeling of the parametric PDEs.

**Questions:**

1.	Some concepts are not explained well. For example, what is Kolmogorov width and why does the decaying of the width cause trouble? What is geometric variation? What is the first-principle mode?  What is MMGP method?
2.	The author should also give references when introducing these concepts.
3.	In the literature review 1.3, the author doesn’t mention the drawbacks of current learning-based methods to solve the problem. Hence, their own design is not well-motivated.
I am not an expert in this domain, and the methodology is hard for me to understand. In general, I have the following questions about the technical details.
4.	What is the functional in Eq. (3) and Eq. (4)? What does this objective mean and why should we introduce them?
5.	In Section 3.1.2, why should we use the Gaussian process regression to estimate the mapping? Did the author test other regression methods?
6.	Section 4 is very short. I doubt that it can be treated as an independent Section. What’s the relation between Sections 3, 4, and 5?
7.	Is solving the Optimization in Eq. (3) (Section 5) your unique contribution? What’s the existing work for this optimization? What are their drawbacks? What’s the novelty for the author’s algorithm?
8.	It seems that the author’s method is based on the so-called mesh morphing Gaussian process (MMGP). However, MMGP is only introduced in the Experiment part (Section 6.4). I suggest that the author to move this part to the Preliminary and Methodology, explain more about the difference and the development from MMGP to O-MMGP.
9.	Finally, the author doesn’t mention about the potential drawbacks and the future work for their methods.
10.	The improvements of the proposed method to the state-of-the-art is not significant.

**Ethical Concerns:**

["NO or VERY MINOR ethics concerns only"]

**Final Justification:**

It seems that the main contribution of the paper is a novel optimization and efficient algorithms to solve the optimization. This optimization seems to be highly relevant to NeurIPS's topic, as it's a nonlinear dimensionality reduction technique. However, I am not an expert in the domain of PDE-based optimization. I can hardly decide on the novelty of this optimization. Hence, I will increase my score to 3 and reduce the confidence score to 1, and let other reviewers decide the final score.

**Limitations:**

The authors need to elaborate on the limitations and future work for their proposed framework. This part is not present in the current manuscript.

**Paper Formatting Concerns:**

The paper exceeds the 9-page limit by 3 lines.

**Quality:**

2

**Strengths And Weaknesses:**

Strength:
1. The author combines optimization with the MMGP to gain improvements over the cutting-edge methods.
2. The author conducts experiments in NeurIPS ML4CFD competition and claims a better result, compared to the wining solution.
3. The author gives thorough proofs for solving the optimization.

Weakness:
In general, the paper is not well-written. Some important concepts are not well explained. The methodology is hard to understand. Please see my following questions in the Question section.

---

> ### Author Rebuttal · Authors · 2025-07-30
>
> We thank the reviewer for all the valualbe remarks.
>
> Response to questions
>
> 1- The Kolmogorov $N$-width measures how well the solution manifold (the set of all solutions to the parametric PDE for all parameter values) can be approximated by an $N$-dimensional linear subspace. It is defined as the smallest worst-case error achievable by any $N$ dimensional subspace of the full solution space. If the Kolmogorov $N$-width decays rapidly as $N$ grows, it means the solution manifold is low-dimensional in a linear sense, and a linear reduced basis of modest size can capture the solutions accurately. However, if the Kolmogorov $N$-width decays slowly with $N$, the solution manifold is inherently “high-dimensional” from the perspective of linear subspaces.
> Generaly, we do not have acces to the explicit expression of the Kolmogorov $N$-width. In order to study the problem, we check the decay of the eigenvalues of the correlation matrix. This is the core idea used in this paper. We find the optimal mappings such that the decay of the eigenvalues of the correlation matrix (thus the Kolmogorov $N$-width) of the transformed solution manifold (after mappings) is reducible, that is can be approximated by a small dimensional subspace.
>
> Geometric variation: this refers to the geometry being different for each sample.
>
> MMGP: it is briefly discussed and referenced in section 6. It refers to the method  where we map each geometry onto a reference geometry using Mesh Morphing, and then we apply dimensionality reduction and the Gaussian Process regression.
>
> 2- We will make sure to reference these concepts and detail them in appendix as needed.
>
> 3- We will mention some drawbacks of the current state of the art. Mainly, our approach is motivated by performing deterministic steps as much as possible before the learning phase. This will make the approach more interpretable from a mathematical point of view. Unlike most deep learning based methods, where the complex architecture make it hard to interpret the results.
> The MMGP method also have many drawbacks that motivate our work. Mainly:
>
> - The morphing is chosen only using geometrical critera and they do not depend on the fields. This would mean that $\phi^{opt}=Id$ the identity map in our methodology.
> - The morphing is contructed manually for each case, unlike the approach here. T
> - The morphing does not tackle the non reducible problem issue.
>
> 4-  The objective function in eq (2) represents the relative energy in the first $r$ component of the POD (proper orthogonal decomposition). This is the analogue of the PCA in infinite dimension. Each sample is approximated on an orthogonal basis of dimension $r$. This objective function can be viewed as the explained variance in the first $r$ directions. If it is close to 1, that means that all the samples can be explained by the $r$ first modes, thus the interest of maximizing this objective function with respect to the mappings.
>
> 5- The use of Gaussian process regression is a choice here, as they provide very good accurate prediction in this context.  They also work very well with a small number of samples in the training set, which is usually encountered in the context of physical simulations, and they provide uncertainty on the results. However, other regression models could be also used. In our experience across various datasets, Gaussian Processes consistently yielded some of the best results, even when compared to tree-based and deep learning regressors.
>
> 6-  Section 3 presents the methodology we introduce in this paper. In this methodology, we have the pretreatment phase which consists of computing $\phi^{geo}$ and $\phi^{opt}$. Section 4 briefly mentions some methods to compute $\phi^{geo}$, as this is an independent problem that is studied in the literature. Section 5 is the core of the article, as it details how to compute $\phi^{opt}$ solution to the introduced objective function.
>
> 7- The main contribution of this work is the introduction of the objective function in eq (2) and the resolution of the maximization problem in eq (3) as detailed in section 5. These elements are all new to the literature. By tackling the challenging issue of non-reducible problems with variable geometry through an optimization approach that maximizes energy in the first principal modes, we directly addresses the core limitation of traditional dimensionality reduction techniques.
>
> The optimization problem in infinite dimension. The resolution scheme proposed in section 5 is also new, giving rigorous mathematical derivation of the differential of the newly introduced objective function. We introduce multiple ideas (section 5.3 and 5.4) that aim to aid the convergence of the algorithm.
>
> 8- Our contribution is compared to the recent winning solution of the ML4CFD NeurIPS 2024 competition, which attracted 240 participating teams and a wide variety of methods, including numerous deep learning approaches such as Implicit Neural Representations, Graph Neural Networks (GNNs), and Transformers. These solutions,particularly the winning method, MMGP,were extensively customized for this specific task.
>
> In the case of MMGP, substantial manual effort was devoted to designing a morphing strategy that aligns turbulence viscosity patterns. This involved a complex mechanism based on an initial learning step to infer the wake angle from the input parameters. While this solution achieved state-of-the-art accuracy, the need for such manual adaptation limits its applicability beyond this benchmark.
>
> The goal of our work is to make this morphing process fully automatic, thereby significantly increasing the practicality and generalizability of MMGP. In our view, the observed accuracy improvement is an additional benefit: despite being automatic, our approach was able to outperform the highly customized winning solution. We will revise the manuscript to explain these points in greater detail.
>
> 9- We briefly mention them in the conclusion. Mainly, we are currently to address the problem when we have curved boundary of the domain (we recall here that we fix the points on the curved boundary).  We will detail them further in the final version. The other major development is testing the method in other context, for example in the presence of different shocks in the dataset samples.
>
> 10 - While the improvement with respect of the state of the art is not very significant, the method introduce a very important concept which is the automatic alignment of the fields by solving the proposed optimization problem without any manual precomputation.

---

> > ### Comment · Reviewer_NSkh · 2025-08-04
> >
> > I would like to thank the authors for the explanations.
> >
> > It seems that the main contribution of the paper is a novel optimization and efficient algorithms to solve the optimization. This optimization seems to be highly relevant to NeurIPS's topic, as it's a nonlinear dimensionality reduction technique. However, I am not an expert in the domain of PDE-based optimization. I can hardly decide on the novelty of this optimization. Hence, I will increase my score to 3 and reduce the confidence score to 1, and let other reviewers decide the final score.

---

### Official Review · Reviewer_MaES · 2025-07-03

**Clarity:** 3
**Significance:** 3
**Originality:** 3
**Rating:** 4
**Confidence:** 2

**Summary:**

The paper introduces O-MMGP, a framework that combines mesh morphing, non-linear dimensionality reduction, and Gaussian Process Regression to address the challenge of efficiently solving parametric PDEs with significant geometric variations and non-reducible problems (e.g., those featuring shocks and moving discontinuities). By defining and optimizing so-called “optimal mappings,” the approach enables a more compact representation of the solution fields, facilitating efficient regression and approximation. The methodology automates alignment and reduces reliance on manual feature tracking, and is empirically validated on both a synthetic advection-reaction PDE and the ML4CFD NeurIPS 2024 airfoil benchmark, with comparison to state-of-the-art and competition-winning methods.

**Questions:**

Scalability to Larger and 3D Problems: Could the authors provide quantitative insight or runtime scaling on how O-MMGP would perform for larger mesh sizes, 3D domains, or higher sample counts? Are there bottlenecks related to the mesh-morphing or the infinite-dimensional optimization that would limit real-world adoption?

Parameter Sensitivity and Robustness: What is the sensitivity of the approach to choices of c1, c2, r, and POD truncation? Would alternative choices substantially degrade performance, or is there a principled way to set them in new problems?

Automated Curved Boundary Handling: As future work is planned to generalize to curved boundaries, are there conceptual or computational obstacles the authors foresee? How critical is this limitation in practice for the domains the authors believe O-MMGP could address?

Uncertainty Quantification: GPR, in general, enables predictive uncertainty. Did the authors make any use of these uncertainty estimates in the downstream tasks or error analysis? If so, could they provide an empirical illustration, for example, on the out-of-distribution (OOD) scenarios?

**Ethical Concerns:**

["NO or VERY MINOR ethics concerns only"]

**Final Justification:**

After considering the rebuttal, I maintain my rating.

**Limitations:**

Yes

**Paper Formatting Concerns:**

Exceed the 9 page limitation.

**Quality:**

3

**Strengths And Weaknesses:**

Strength:
1. Technical Strength and Theoretical Foundation: The formulation of the optimal mapping problem is mathematically well-grounded, providing a rigorous optimization framework for enhancing the compressibility of solution fields with slowly decaying Kolmogorov width. The use of an explicitly derived functional and an infinite-dimensional gradient algorithm shows solid theoretical backing.
2. Meaningful Advancement over State-of-the-Art: The approach enables automatic, principled multi-modal alignment for non-parametric geometry PDEs, which traditionally suffer from high-dimensional irreducibility when shocks or discontinuities move.
3. Empirical Rigor: The paper conducts comprehensive experiments on both a synthetic advection-reaction PDE (Figure 2) and the widely adopted AirfRANS/ML4CFD dataset (Section 6.2), comparing the proposed O-MMGP to both original MMGP and the NeurIPS 2024 competition winner.

Weakness:
1. Experimental Scope and Diversity: The experimental validation, while strong in its target domains, focuses primarily on two examples: a synthetic 2D PDE and the ML4CFD/Gaussian Process airfoil data. No evidence is provided for higher-dimensional problems (e.g., 3D shapes) or alternative domains, limiting claims of broad generalization.
2. Baseline and Ablation Details: The paper compares O-MMGP against MMGP and one expert-tuned baseline (the NeurIPS winner), but misses some possible baselines, e.g., recent neural operator methods or advanced geometric deep learning surrogates for variable domains [such as Geo-FNO, DeepONet in more detail]. The ablation with respect to various components (optimal mapping, field transformation, regression model choice) could be fleshed out (Section 6.4 references parameter choices but does not deeply analyze their impact).
3. Parameter Sensitivity and Hyperparameter Selection: While the paper specifies values for key parameters (e.g., c1, c2, r), it does not systematically analyze sensitivity or robustness to these choices. The effect of continuation parameters and the impact of the elastic penalty for mapping bijectivity are mentioned but not quantitatively explored (see Section 5.2, and empirical discussion).
4. Scalability and Computational Cost: The method involves potentially expensive optimization in infinite-dimensional function spaces, with multiple iterative or continuation steps. While the cost is referred to (see Section 6.4, and appendix references), there is limited main-text quantification of run-time or resource needs for larger-scale or more complex problems, and the impact of mesh size or sample count is not discussed.

---

> ### Author Rebuttal · Authors · 2025-07-29
>
> We thank the reviewer for all the valuable remarks.
>
> Weaknesses:
>
> 1- The method is proposed to problems with polytope domains. We validated the method in two problems with rectangular domains (in the second one, we fixed the nodes on the boundary), which corresponds well with the hypothesis of the method. From a theoretical point of view, while we only tested the method for $d=2$, it is actually agnostic of the dimension. From a practical point of view, we are more vulnerable to the bottlenecks mentioned in appendix D (see below for further details).
>
> 2- Since we are comparing with the Neurips 2024 winning solution, we are also comparing with all the state of the art approaches that were also used in the competition ( see [2]). Indeed comparing to other approaches such as Geo-FNO and DeepONet would have been useful in this context also. The regression model used here was fixed to compare the effect of the optimal mapping on the learning phase. Other well tuned regression model would have the same effect, since by the proposed method, we are compressing the solution manifold better. Internal tests illustrated that morphing pretreatment enabled 30% RMSE improvements of vision transformer-based methods.
>
> 3- We discussed the sensitivity of the initial choices of $c_1$ and $c_2$ in appendix C.3, repeated here for reference:
>  "In our numerical examples, we find that the method is not sensitive for the initial choices for both parameters as expected. In fact, choosing a large value for $c_1$ would force the mapping to be close to the identity map. In this case, the mesh would not deform by much and the algorithm would converge quickly, and in return we change the value of $c_1$. On the other hand, choosing a small value of $c_2$  would have a similar effect in terms of convergence. A small value would diffuse very much the values of the field in the domain, and they approach a constant field allowing to converge rapidly and to change the values of $c_2$ in return."
>
> Number of modes $r$ : the most delicate parameter between the three. If we do not have any intuition about the shape of the solutions, we should try different number of modes. Less modes is better for learning. Bigger number of modes: less mesh deformation thus decrease the risk of having non bijective mappings, and smaller optimization time (in the extreme case of $r=n$, the energy is maximal, so the identity is an optimal solution, the reference mesh does not deform and no optimization is required), but the prediction becomes much less accurate for non reducible problems. For both cases considered in the numerical tests, we choose $r=1$. We could also report the solution of the mappings for $r>1$ if necessary, as we consider this is an advantage in our method with other similar registration methods in the literature.
>
> Questions
>
> 1- Scalability : Dimension of the problem: Suppose that we have a regular grid in a $d$-dimensional space, and we sample 10 points per dimension, we need $\mathcal N= 10^d$ nodes. The (sparse) linear system to solve would be of dimension $\mathcal N \times  \mathcal N$ (the  $\mathcal N \times  \mathcal N$ matrix is factorized once in offline). The number of integration points would also grow proportionally to the number of nodes. Furthermore, evaluating $u_i$ at the increasing number of integration points would be more costly. However, the above limitations are handled very well for problems of dimensions $d=2,3$ in our codes. We should however test the method on a 3d problem to verify these claims.
>
> The cost is mainly determined by the size of the reference mesh and the number of samples. To mitigate the problem related to the reference mesh, we chose a coarse reference mesh to calculate the mappings, then evaluated the mappings on a finer mesh after solving the optimization problem. We did not try any strategies to solve the problem of the high number of samples, but grouping similar samples and using a reduced-order basis based on a dictionary could mitigate this problem by solving several smaller optimization problems.
>
>
> 2- Parameter Sensitivity and Robustness: see remark 3 above.
>
> 3- Automated Curved Boundary Handling: this is always a work in progress. Conceptually, points on the boundary should always glide on the boundary, which could be done by changing the inner product $a$ mentioned in section 5. The computational obstacle is that this operator should be updated at each iteration, which will increase training time. However, the methodology might remains the same. This limitation greatly depends on the application: for example, in the case of a moving shock on an airfoil, imposing zero displacement on the airfoil might not be sufficient.
>
> 4- The predictive uncertainty provided by the variance of the Gaussian Processes (GPs) has been thoroughly addressed in the original MMGP paper [1]. In particular:
>
> - Table 2 reports standard deviations for both relative RMSE and Q² across all experiments.
> - Figure 4 illustrates the predicted output variance for the pressure field: the uncertainty is higher in regions where the error is also larger.
> - Figure 5 shows 95% confidence intervals with respect to a scalar input parameter; the intervals widen appropriately as the input moves away from the training range, demonstrating discriminative uncertainty quantification.
> - The Prediction Interval Coverage Probability (PICP) at the 95% level was computed as 93.05% and 93.5% for two scalar outputs in the AirfRANS experiment, confirming the reliability of the predictive intervals.
> - Figures 15 to 20 provide additional qualitative assessments: they show predicted values, GP variances, and errors. The variance is well correlated with the observed error; the reference solution lies within the confidence intervals for in-distribution test samples, while this property degrades—as expected—on out-of-distribution (OOD) data. The confidence intervals are also well calibrated: tight on the training set, wider on in-distribution test data, and substantially wider OOD.
>
> Given that uncertainty quantification was already systematically evaluated in [1], the present paper deliberately focuses on a complementary aspect: the optimal and automated construction of morphing maps. Our goal is to advance MMGP's practical deployment by addressing efficiency, generality, and usability, while considering the predictive uncertainty aspects as already well established in prior work. We will make this scope more explicit in the revision and direct readers to the original paper for details on predictive uncertainty.
>
> [1] Casenave, Fabien, Brian Staber, and Xavier Roynard. MMGP: a mesh morphing Gaussian process-based machine learning method for regression of physical problems under nonparametrized geometrical variability. Advances in Neural Information Processing Systems 36 (2023): 43972–43999.
>
> [2] Yagoubi, Mouadh, et al. "NeurIPS 2024 ML4CFD Competition: Results and Retrospective Analysis." arXiv preprint arXiv:2506.08516 (2025).

---

### Official Review · Reviewer_sbkm · 2025-07-03

**Clarity:** 4
**Significance:** 3
**Originality:** 3
**Rating:** 4
**Confidence:** 4

**Summary:**

This paper presents a novel nonlinear dimensionality reduction technique for solving parametric PDEs with non-parametrized geometric variations. Combining with morphing techniques, non-linear dimensionality reduction, and Gaussian Process Regression, this approach enables efficient and accurate modeling of complex parametric PDEs defined on variable geometries. The experimental results demonstrate the efficiency performance comparable to state-of-the-art methods, including the winning solution of the ML4CFD NeurIPS 2024 competition.

**Questions:**

Could you elaborate on the specific implementation details of the geometrical mapping?

**Ethical Concerns:**

["NO or VERY MINOR ethics concerns only"]

**Limitations:**

Yes.

**Paper Formatting Concerns:**

There misses Xlabel in Fig5.

**Quality:**

3

**Strengths And Weaknesses:**

Strengths:
By introducing nonlinear dimensionality reduction, the model contains the three approximations parts, which maintains the geometric transformation from reference configuration to the optimal configuration and couples with field-dependent optimization. This method combines two Gaussian process regression.
The novelty of the algorithm lies in the fact that it automatically aligns the functions on an arbitrary number of modes.
Weaknesses：
This paper will be better if developing a more rigorous mathematical analysis of the approach.

---

> ### Author Rebuttal · Authors · 2025-07-29
>
> We thank the reviewer for their comments.
>
> Geometric mapping is performed using the RBF morphing technique. We begin by selecting control points, which are nodes on the airfoil profile. We then map each control point of the reference domain airfoil onto the target domain airfoil. The position of the mapped point follows the curvilinear abscissa on the airfoil. Once the control points have been mapped, we apply RBF interpolation to each component of the mapping in order to obtain the mapping over the entire domain, as mentioned in Appendix E.

---

### Note · Authors · 2025-08-13

We thank the reviewers and AC for their time, constructive feedback, and valuable suggestions, which have helped us strengthen the manuscript.
Our work introduces a novel optimal mesh morphing Gaussian process framework (O-MMGP) that automates the alignment of solution fields across non-parametric geometric variations, a long-standing challenge in model order reduction for PDEs, making previously non-reducible problems accessible. The core innovation is solving an infinite-dimensional optimization problem to obtain optimal mappings. It significantly enhances the compressibility of solution manifolds, enabling accurate prediction even for geometries that cannot be handled by standard POD-based approaches.
We addressed reviewer concerns on:

- Scope and generality: While our experiments target two representative domains (synthetic advection-reaction PDE and the ML4CFD competition), the methodology is dimension-agnostic and applicable to other PDEs, including in 3D.
- Regression model: Gaussian process regression consistently yielded some of the best results in our experiments when compared to other regression models tree-based and deep learning regressors. However, other regression models could be equally tested.
- Relation to other methods: Our mapping could be used also as a preprocessing step for operator-learning methods (FNO, DeepONet, Geo-FNO). We tested coupling with vision transformer architecture, which yielded significant RMSE reductions in early experiments.
- Robustness: The bijectivity of mappings is actively monitored and enforced; failure cases trigger early stopping. Sensitivity to parameters ($c_1, c_2, r$) was tested, showing stable results across reasonable ranges.
- Computational cost: The optimization step is a one-time offline cost. Coarse to fine mapping reduces runtime.
- Uncertainty quantification: Predictive uncertainty was extensively evaluated in prior MMGP work; here, we focus on advancing automation and usability while preserving well-calibrated uncertainty estimates.

In summary, O-MMGP offers a principled, automatic, and extensible approach that couples morphings pretreatment with regression, improving predictive accuracy while reducing manual intervention. We believe these contributions, together with the methodological rigor and empirical validation, make this work a valuable addition to the conference program.

---

### Decision · Program_Chairs · 2025-09-17

**Decision:**

Reject

**Comment:**

This paper proposes O-MMGP, an extension of Mesh Morphing Gaussian Process regression to handle non-parametric geometric variations in PDEs via an infinite-dimensional optimization for “optimal mappings.” The method is well-motivated and technically sound, and in principle it is on-topic for NeurIPS.

However, the reviews are mixed: two reviewers could not make much of the paper, one senior reviewer gives a clear and well-argued reject, and no reviewer could give an unqualified acceptance statement. Concerns include the very narrow experimental scope (only one synthetic and one 2D benchmark), limited baseline comparisons, lack of robustness or scalability studies, and dense, jargon-heavy presentation. While I could follow the work—given my math background and its proximity to my research—the writing is highly technical in PDE theory, numerics, and related areas, making it hard for a broader NeurIPS audience to engage with.

While the paper is on-topic, it seems unlikely to find a substantial audience at NeurIPS in its current form. The authors might either consider submitting to a venue more focused on numerical PDE methods, or reworking the presentation into a more accessible, machine-learning style with expanded experimental validation.